# COMBINING MIXMATCH AND ACTIVE LEARNING FOR BETTER ACCURACY WITH FEWER LABELS

## ABSTRACT

We propose using active learning based techniques to further improve the state-of-the-art semi-supervised learning MixMatch algorithm. We provide a thorough empirical evaluation of several active-learning and baseline methods, which successfully demonstrate a significant improvement on the benchmark CIFAR-10, CIFAR-100, and SVHN datasets (as much as 1.5% in absolute accuracy). We also provide an empirical analysis of the cost trade-off between incrementally gathering more labeled versus unlabeled data. This analysis can be used to measure the relative value of labeled/unlabeled data at different points of the learning curve, where we find that although the incremental value of labeled data can be as much as 20x that of unlabeled, it quickly diminishes to less than 3x once more than 2,000 labeled example are observed.

## 1 INTRODUCTION

Sophisticated machine learning models have demonstrated state-of-the-art performance across many different domains, such as vision, audio, and text. However, to train these models one often needs access to very large amounts of labeled data, which can be costly to produce. Consider, for example, laborious tasks such as image annotation, audio transcription, or natural language part-of-speech tagging. Several lines of work in machine learning take this cost into account and attempt to reduce the dependence on large quantities of labeled data. In *semi-supervised learning* (SSL), both labeled and unlabeled data (which is often much cheaper to obtain) are leveraged to train a model. Unlabeled data can be used to learn properties of the distribution of features, which then allow for more sophisticated and effective regularization schemes. For example, enforcing that examples close in feature space are labeled similarly. Another, different, approach for address costly labeled data is that of *active learning* (AL). Here, a model is still trained using only labeled data, but extra care is taken when deciding which unlabeled data examples are to be labeled. Often, the data will be labeled iteratively in batches, where at each iteration an update is made to a current view of the distribution over labels and the next batch of points is selected from regions where the distribution is least certain. As discussed in depth in the following section, the approaches of semi-supervised and active learning can be complementary and used in conjunction to help solve the problem of costly in labels.

**Our Contributions:** In this paper, we take MixMatch, the leading semi-supervised learning technique, and thoroughly evaluate its performance when combined with several active learning methods. We find very encouraging results, which show that state-of-the-art performance is achieved in the limited label setting on CIFAR-10, CIFAR-100, and SVHN datasets, demonstrating that combining active learning techniques with MixMatch provides a significant improvement. Furthermore, we perform an analysis, exploring the incremental benefits of labeled versus unlabeled data at different points in the learning curve. Given the relative costs of labeled and unlabeled data, such an analysis aids us in deciding how to best spend a given budget in acquiring labeled versus unlabeled data.

The remainder of the paper is organized as follows: We first give a high-level review of active learning related work and the MixMatch algorithm in Sections 2 and 3, respectively. The evaluated methods and experimental setup are described in Section 4, while the experimental results are presented and analyzed in Section 5. Finally, we conclude in Section 6.

## 2 ACTIVE LEARNING AND RELATED WORK

Active learning (or active sampling) methods are designed to answer the question: *Given a limited labeling budget, which examples should I expend my budget on?* Like semi-supervised learning, active learning is particularly is useful when labeled data is costly or scarce for any host of reasons. Generally, an active learning algorithm iteratively selects samples to label, based on the data labeled thus far as well as the model family being trained. Once the labels are received, they are added to the labeled training set and the process continues to the next iteration, until the budget is finally exhausted. Thus, active learning can be considered to be part of the model training process, where classical model training is interleaved with data labeling. Here we give a very brief introduction to a few classes of active learning algorithms, and their use with semi-supervised learning.

There are several active learning algorithms with strong theoretical guarantees, which either explicitly or implicitly define a *version space* that maintains the set of "good" candidate classifiers based on the data labeled thus far. The algorithms then suggest labeling points that would most quickly shrink this version space (Cohn et al., 1994; Dasgupta et al., 2008; Beygelzimer et al., 2009). In practice, tracking this version space can be computationally inefficient for all but the simplest (e.g. linear) model families. Another, related approach, is that of *query-by-committee*, where points are selected for labeling based on the level of disagreement between a committee of classifiers, which have been selected using the currently labeled pool. Query-by-committee methods benefit from theoretical guarantees (Abe & Mamitsuka, 1998), as well as practical implementations based on bagging (Seung et al., 1992). These methods, along with uncertainty sampling that we introduce next, are only a few prototypical examples of active learning algorithms. For a broader introductory survey of algorithms and techniques, please see Settles (2009) and the references therein.

Arguably, the most popular and practically effective active learning technique is that of *uncertainty* or *margin* sampling (Lewis & Gale, 1994; Lewis & Catlett, 1994). At each iteration, this approach first trains a model using the currently labeled subset, then it makes predictions on all unlabeled points under consideration and, finally, it queries labels for those points where the confidence in the models' prediction is the smallest. The notion of confidence can be defined several different ways, as will be detailed in Section 4.2. In this light, we can see that active learning methods and semi-supervised learning methods can work in a complementary fashion. At a high level, active learning methods seek to find training examples for which we have least confidence in the underlying labels in order to query for those labels, while semi-supervised learning algorithms can focus on training examples where there is strong confidence in the label distribution and reasonable assumption conclusions can be made regarding unlabeled examples. It is no surprise that combining active learning and semi-supervised learning has been investigated previously (Hoi et al., 2009; Muslea et al., 2002). We highlight a few of these examples below (also see Section 7.1 of Settles (2009)).

One of the earliest lines of work combining the two techniques is that of McCallum & Nigam (1998), where the query-by-committee framework is combined with an expectation maximization (EM) approach that is used to provide pseudo-labels to those examples that have not been queried for true underlying labels by the active learning method. They show that, on a text classification task, the improvement in label complexity of the combined method is better than what either of semi-supervised or active learning method alone can provide. In the work of Zhu et al. (2003), the two approaches are combined using a Gaussian random field model. Given a model trained on the union of currently labeled and unlabeled datasets, the expected reduction in the estimated risk due to receiving the label of a particular example can be computed and greedily optimized. Tur et al. (2005) and Tomanek & Hahn (2009) both combine uncertainty sampling to label uncertain points with machine labeling of confident examples and show their effectiveness in applications to a spoken language understanding and sequence labeling tasks, respectively. In this work, we combine uncertainty-sampling based active learning method and demonstrate that the already state-of-the-art semi-supervised performance of the MixMatch algorithm can be significantly further improved.

## 3 MIXMATCH

We only focus on deep learning based semi-supervised learning (SSL) in this work. We use the framework from Oliver et al. (2018) to perform a realistic evaluation of the results (same splits of the labeled/unlabeled initial data, same network architecture). Recent popular techniques are built

around two concepts: Consistency Regularization (Mean Teacher (Tarvainen & Valpola, 2017), Π-model (Laine & Aila, 2017),Virtual Adversarial Training (VAT) (Miyato et al., 2018)) which enforces that two augmentations of the same image must have the same label and Entropy Minimization (Pseudo-Label Lee (2013), Entropy Minimization (EntMin) (Grandvalet & Bengio, 2005)) which states that a prediction should be confident. These two concepts are sometimes combined, for example the technique VAT EntMin is a combination of VAT and Entropy Minimization and MixMatch is also using both of these concepts.

MixMatch, a recent state-of-the-art semi-supervised learning (SSL) technique, is designed around the idea of guessing labels for the unlabeled data followed by using standard fully supervised training (Berthelot et al., 2019). Consider a classification task with classes $\mathcal{C}$. The input of MixMatch are a batch of $B$ (images, labels) pairs $X = \{(x_b, p_b)\}_{1 \le b \le B}$, where each label $p_b$ is an one-hot vector over the class $\mathcal{C}$, and a batch of unlabeled examples (images) $U = \{u_b\}_{1 \le b \le B}$. Note, in this section, a "batch" is in reference to the subset of points used in the iterative optimization procedure that is used to train the model, while elsewhere in the paper "batch" refers to a set of newly labeled points that are added to the training set during the iterative active learning process.

**Label Guessing.** Label guessing is done by averaging the training model's own prediction on several augmentation $\hat{u}_{b,i}$ of a same image $u_b$. This average prediction is then sharpened to produce a low-entropy soft label $q_b$ for each image $u_b$. Formally, the average is defined as $\bar{q}_b = \frac{1}{K} \sum_{i=1}^{K} p_{\text{model}}(y|\hat{u}_{b,k}; \theta)$ where $p_{\text{model}}(y|x; \theta)$ is the model's output distribution over class labels $y$ on input $x$ with parameters $\theta$. A sharpening is applied to the average prediction: $q_b = \text{Sharpen}(\bar{q}_b)$. In practice, MixMatch uses a standard softmax temperature reduction computed as $\text{Sharpen}(p)_i := p_i^{1/T} / \sum_{j=1}^{|\mathcal{C}|} p_j^{1/T}$ where $T = 1/2$ is a fixed hyper-parameter.

**Data Augmentation.** MixMatch only uses standard (weak) augmentations. For the SVHN dataset, we use only random pixel shifts, while for CIFAR-10 and CIFAR-100 we also use random mirroring.

**Fully supervised.** Finally, MixMatch uses fully supervised techniques. In practice, it use weight decay and MixUp across the labeled and unlabeled data (Zhang et al., 2017). In essence, MixUp generates new training examples by computing the convex combination of two existing ones. Specifically, it does a pixel level interpolation between images and pairwise interpolation between probability distribution. The resulting interpolated label is a soft label. Such examples encourage the model to make smooth transitions between classes. Let $\hat{X} = \{(\hat{x}_b, p_b)\}_{1 \le b \le B}$ and $\hat{U} = \{(u_b, q_b)\}_{1 \le b \le B}$ be the results of data augmentation and label guessing. MixMatch shuffles the union of the two batches $\hat{X} \cup \hat{U}$ into a batch $W$ of size $2B$ and performs MixUp to produce the output:

- $X' = \text{MixUp}(\hat{X}, W_{[1,...,B]})$ (mixing up $\hat{X}$ with the first half of $W$) and
- $U' = \text{MixUp}(\hat{U}, W_{[B+1,...,2B]})$ (mixing up $\hat{U}$ with the second half of $W$).

Given two examples $(x_1, p_1)$ and $(x_2, p_2)$ where $x_1$, $x_2$ are feature vectors and $p_1$, $p_2$ are one-hot encoding or soft label, depending on whether the corresponding feature vector is labeled or not, MixMach performs MixUp as follows:

(1) Sample $\lambda \sim \text{Beta}(\alpha, \alpha)$ from a Beta distribution parameterized by the hyper-parameter $\alpha$;
(2) For $\lambda' = \max(1 - \lambda, \lambda)$, compute $x' = \lambda' x_1 + (1 - \lambda') x_2$ and $p' = \lambda' p_1 + (1 - \lambda') p_2$.

**Loss function.** Similar to other SSL paradigms, the loss function of MixMatch consists of a sum of two terms: (1) a cross-entropy loss between a predicted label distribution with the ground-truth label and (2) a Brier score (L2 loss) for the unlabeled data which is less sensitive to incorrectly predicted labels. On a MixMatch batch $(X', U')$, the loss function is $\mathcal{L} = \mathcal{L}_X + \lambda_U \mathcal{L}_U$ where

$$\mathcal{L}_X = \frac{1}{|X'|} \sum_{(x,p) \in X'} \text{CrossEntropy}(p, p_{\text{model}}(y|x; \theta)), \tag{1}$$

$$\mathcal{L}_U = \frac{1}{|\mathcal{C}||U'|} \sum_{(u,q) \in U'} \|q - p_{\text{model}}(y|u; \theta)\|_2. \tag{2}$$

Here $\lambda_U$ is the hyper-parameter controlling the importance of the unlabeled data to the training process. We set $\lambda_U$ to be 75 for CIFAR-10, 150 for CIFAR-100, and 250 for SVHN. We also fix the MixUp hyper-parameter $\alpha = 0.75$.

## 4 Proposed Method and Analysis

In this section, we describe our proposed method for introducing active learning into MixMatch, we then review several specific active learning strategies which we will be evaluated, and finally we also describe our methodology for analyzing the relative value of labeled versus unlabeled data.

### 4.1 Proposed Method

In the standard semi-supervised learning setting, a classifier is trained with $n$ samples where only $m$ of them are labeled ($m \ll n$). The $m$ labeled samples are usually considered to be a fixed and uniformly sampled subset from the $n$ samples. This is also the setting that the original MixMatch algorithm considers (Berthelot et al., 2019). In this section, we propose `MMA`, a combination of MixMatch and active learning as well as define a method for comparing the relative value of labeled and unlabeled samples.

**Direct gains from active learning** A natural extension is to consider whether using *active learning* in place of uniform sampling can improve MixMatch results. In practice, instead of randomly sampling $m$ samples to be labeled all at once, we incrementally grow the labeled set as the training process goes on. Starting with a fixed pool of $n$ unlabeled sample, we first randomly sample a small labeled set of $L_0$ of size $m_0$. This set is grown k times as the training progresses $L_0 \subset L_1 \subset \ldots \subset L_k$ with respective sizes $m_0 < m_1 < \ldots m_k$. The selection process to determine what labels $L_{i-1} - L_i$ to add at step $i$ is done using active learning. For example, from a fixed set of $n$ samples, we start with a set $L_0$ formed by labeling 250 randomly selected samples. After training for some time, we can grow by 50 unlabeled examples that the current model considers "hardest" (i.e., the most uncertain ones).

**Determining the worth of labels** Both labeled data and unlabeled data have a cost. It is commonly the case that labeled data is much more expensive than unlabeled data. From a joint active learning and semi-supervised learning perspective, two datasets can be grown: the labeled set L and the unlabeled set U. In this context, the question to be answered is this: with the goal to reach a target accuracy $accuracy_{target}$, at a time step $t$ is it better to grow the labeled set or the unlabeled set? As a corollary question, how do various accuracy targets relate to each other? In other words, can the model response to data be predicted from lower accuracy targets?

### 4.2 Comparison of Popular AL Methods

It is a common strategy in active learning to select the samples that the current model, trained on the data that is available thus far, is most uncertain about. However, directly selecting the most uncertain samples can result in a large number of similar samples. Existing methods avoid this issue by picking uncertain yet diverse samples. See Settles (2009) for a survey of these and several other active learning techniques. The design of `MMA`'s active learning strategies considers two components: (1) uncertainty measure and (2) diversification.

**Measuring uncertainty.** Following previous active learning frameworks, `MMA` gathers labels of the samples that the current model is most uncertain about. Intuitively, the uncertain samples are often the most helpful for improving the model. The first question we ask is how should we measure uncertainty? We consider two approaches.

For a sample $x$, the current model predicts its label as a probability vector $p_{\text{model}}(y|x;\theta) \in [0,1]^{|\mathcal{C}|}$ where the $c$-th dimension $p_{\text{model}}(y|x;\theta)_c$ is the probability of assigning $x$ to class $c$. Let $s(x)$ be the uncertainty of sample $x$. Common ways of measuring uncertainty include

- `max`: Measuring the maximum confidence that the model has in any one label $s(x) := 1 - \max_c\{p_{\text{model}}(y|x;\theta)_c\}$ .
- `diff2`: Measuring the gap margin between the two most likely classes, $s(x) := 1 - (p_{\text{model}}(y|x;\theta)_{c_1} - p_{\text{model}}(y|x;\theta)_{c_2}(x))$, where $c_1$ and $c_2$ are the classes with the 1st and 2nd highest probabilities.

Additionally, we can reuse a MixMatch technique to make the uncertainty measurement more robust. Instead of using the model prediction on the original sample, we can average its predictions from $K$ different augmented versions of the example. We call this method `aug` and combine it with `max`

and `diff2`. It results in two more measurement methods: `max.aug` and `diff2.aug`. We choose $K = 2$ to stay consistent with the setting of MixMatch.

**Diversification.** Always simply sampling the most uncertain samples (which we will call the `direct` method) can cause the problems that the sampling targets specific subsets of the data (typically those larger classes). In such cases, it is helpful to add a form of regularization to ensure diversity within our sampled batch. `MMA` considers two commonly used diversification methods:

- `kmeans`: We first cluster all unlabeled samples using the k-means clustering algorithm (Lloyd, 1982). At each sampling step, we select the top-$n$ uncertain samples from each cluster where the sample size $n$ is proportional to the cluster size. We fix the number of clusters to be 20.
- `infoD`: the information density framework adds an additional term that measures how representative the sample is, i.e., for a sample $x$, we calculate $s'(x) = s(x) \times \left( \frac{1}{|\mathcal{U}|} \sum_{x' \in \mathcal{U}} \mathrm{sim}(x, x') \right)^{\beta}$ for some uncertainty measure $s(\cdot)$, similarity measure $\mathrm{sim}(\cdot, \cdot)$ and some user-defined value $\beta$. Note, if $|\mathcal{U}|$ is prohibitively large, $s'(\cdot)$ can be computed using a uniform random subsample. The `infoD` method picks the samples with highest $s'$ values. In our experiment, we set $\beta = 1$ and set $\mathrm{sim}(\cdot, \cdot)$ to be the cosine similarity.

In both methods, we need some representation of samples to measure the distance or similarity of samples. Instead of using the original feature vector, in our experiments, we use the output the second from the last layer of the neural network as the embedding representation of the input.

**Other active learning methods.** Although other families of active learning algorithms exist (such as those found in Settles (2009)), we find that they are not easily adapted to this setting complex and costly to train neural networks. Ensemble-based methods, such as query-by-committee (Seung et al., 1992), require training many copies of a model resulting in a resource and potentially time bottle-neck. Version-space based approaches, such as IWAL (Beygelzimer et al., 2009) or DHM (Dasgupta et al., 2008) are not readily applicable in a setting with a complex hypothesis space, where either explicitly or implicitly tracking the version space becomes a complex optimization problem itself. Perhaps expected model change (Settles et al., 2008) or expected error reduction (Roy & McCallum, 2001) are the baseline methods that are the next most amenable to the setting, however, even in these cases one would need to compute gradients/predictions for each individual example and each possible labeling. In cases where the number of labels is large, e.g. CIFAR-100, the direct use of such methods become impractical.

## 4.3 Cost Analysis Model for Labeled vs Unlabeled data

In this section, we formalize how we propose to perform the cost analysis of adding labeled vs unlabeled data. Let $c_l$ and $c_u$ be the costs of obtaining a new labeled sample and a new unlabeled sample respectively. For desired accuracy $accuracy_{target}$, we can obtain several groups of (labeled, unlabeled) samples that allow the trained model to reach this accuracy. In a realistic setting, as long as $accuracy_{target}$ is less than the model best accuracy, such groups can be created by removing data from $U$ and $L$ until the model reaches the desired accuracy. Formally, let's consider an ordered series of training sets $T_{i,j} = (L_i, U_{i,j})$ where $\forall i < k, L_i \subset L_k$ and $\forall j < h, U_{i,j} \subset U_{i,h}$. Given two contiguous training sets $T_{i,j}$ and $T_{i+1,k}$ reaching $accuracy_{target}$, we can estimate under what condition it is better to add labeled data by solving: $c_l|L_i| + c_u|U_{i,j}| > c_l|L_{i+1}| + c_u|U_{i+1,k}|$. In doing so, we obtain the cost ratio $c_{ratio(i)}$ of unlabeled data over labeled data:

$$c_{ratio(i)} = (|U_{i,j}| - |U_{i+1,k}|) / (|L_{i+1}| - |L_i|).$$

If $c_{ratio(i)} > c_l/c_u$, we can conclude it is better to collect labeled data and otherwise it is better to collected unlabeled data.

## 5 Experiments

In this section, we evaluate the effectiveness of `MMA` by applying it to image classification tasks under the standard semi-supervised and active learning setting. We compare the performance of different active learning variants of `MMA` with the non-active MixMatch. As an additional contribution, we conduct an extensive cost analysis of active data labeling under varied dataset sizes.

## 5.1 DATASETS, EXPERIMENT SETTINGS, AND IMPLEMENTATION DETAILS

We evaluate the effectiveness of `MMA` on three semi-supervised learning benchmarks: CIFAR-10, CIFAR-100 (Krizhevsky et al. (2009)), and SVHN (Netzer et al. (2011)). CIFAR-10 and CIFAR-100 each consists of 50000 images classified into 10 and 100 classes. SVHN consists of 73257 images of street view house number, classified into 10 classes. An extra set of 531131 samples exists for SVHN; we call the combination of the two, which is a dataset with 604388 examples, SVHN+Extra.

Our implementation is primarily based on that of MixMatch (see Berthelot et al. (2019) and GitHub repo referenced therein). The implementation uses the wide ResNet-28 model in Oliver et al. (2018). We also utilize weight decay and exponential moving averaging, again following the settings in MixMatch. All hyper-parameters related to MixMatch are the same as in (Berthelot et al., 2019). Details can be found in Appendix A.1. We report the median of the last 20 checkpoints' accuracy where a checkpoint is computed every 1024 training iterations. Each experiment is repeated 5 times with different random initial sets, and we report the mean and standard deviation.

Each run of the experiment is given a fixed labeling budget. The training process proceeds as follows. `MMA` first starts training the model with the initial labeled examples. After a fixed number of training steps, `MMA` grows the labeled set by querying the labels of some unlabeled examples; this is repeated until the labeling budget is used up. Finally, the model is trained futher until convergence.

To save computational resources, when we vary the budget size for the same active learning algorithm, we store the model checkpoint at the end of each labeling interval and resume from it when we increase the budget size.

**Baselines:** We compare `MMA` with two baselines: `MixMatch` and `random`. `MixMatch` essentially considers the passive setting: the set of labeled data is sampled uniformly at random in the beginning and is fixed throughout the entire training process. The `random` baseline is under the `MMA` framework, and selects randomly as set of samples to query each time.

## 5.2 EFFECTIVENESS OF MMA

We test the effectiveness of the different active learning strategies used in `MMA`. Recall that `MMA` considers two uncertainty measurements: `diff2` and `max`. Each has its data augmentation variant which we denote as `diff2.aug` and `max.aug`. Moreover, we consider using the uncertainty measurement directly (`direct`) and with diversification strategies `infoD` and `kmeans`. Due to space constraints, we present the accuracy of a subset of representative AL methods (`diff2.aug-direct` and `diff2.aug-kmeans`) which are consistently the best performers (see comparison with `MMA` (best) in Tables 1, 2, and 3 with full tables available in Appendix A.2.

Several of the `MMA` variants improve significantly upon MixMatch, which is the state-of-the-art semi-supervised learning algorithm. With the same labeling budget `MMA` outperforms the original MixMatch by up to 1.47% for CIFAR-10, by up to 1.16% for CIFAR-100, and by 0.43% for SVHN.

**CIFAR-10.** We evaluate `MixMatch` and all variants of `MMA` on CIFAR-10 with 500, 1000, 2000 and 4000 labeling budget. We first train the model for 262144 steps with 250 randomly selected labeled samples. Then we grow the labeled set by 50 examples at each active learning iteration. Each time after obtaining more labeled samples, we continue training the model for 32768 steps.

Table 1: Performance of `MMA` on 5 repeated runs on CIFAR-10.

| Label Budgets | 500 | 1000 | 2000 | 4000 | 16000 |
|---|---|---|---|---|---|
| `MixMatch` | $90.58 \pm 0.83$ | $91.61 \pm 0.54$ | $93.20 \pm 0.11$ | $93.70 \pm 0.16$ | $94.99 \pm 0.15$ |
| `random` | $90.54 \pm 0.82$ | $91.85 \pm 0.67$ | $92.97 \pm 0.29$ | $93.73 \pm 0.18$ | - |
| `diff2.aug-direct` | $91.69 \pm 0.52$ | $92.79 \pm 0.41$ | $94.11 \pm 0.14$ | $95.17 \pm 0.13$ | - |
| `diff2.aug-kmeans` | $91.46 \pm 0.38$ | $92.62 \pm 0.32$ | $93.98 \pm 0.14$ | $95.06 \pm 0.19$ | - |

Table 1 shows the results for CIFAR-10. All the active learning methods outperforms the `random` or the `MixMatch` baselines (see details from Table 5 in Appendix A.1). The improvement is consistently about 1% across different label budgets. It is clear that acquiring labels actively is beneficial

in the semi-supervised setting. MMA is also significantly more *label-efficient*. Even with only 2000 labels, MMA outperforms MixMatch with 4000 ($2\times$) labels (94.11% vs. 93.7%). With 4000 labels, MMA outperforms MixMatch with 16000 ($4\times$) labels (95.17% vs. 94.99%). This is only 0.66% lower than the same model trained on all the 50000 ($12.5\times$) examples in the supervised setting (95.17% vs. 95.83%, number is from Berthelot et al. (2019)).

Comparing the different active learning strategies, we found that diff2.aug-direct consistently ranks top-2 among all active learning variants for CIFAR-10, with other active learning strategies performing roughly equally well. Additionally, we observe that diff2 outperforms max on average and it is usually beneficial to use aug (see Tabel 5 in Appendix A.2). We therefor consider only diff2 as the uncertainty meaurement for the other datasets.

**CIFAR-100.** We evaluate MixMatch and variants of MMA that uses diff2 as uncertainty measurement on CIFAR-100 with 4000, 5000, 8000 and 10000 labeling budget. We first train the model for 262144 steps with 2500 randomly selected labeled samples. Then we grow the labeled set by 500 examples at each active learning iteration and train the model for 32768 additional steps. Note that since CIFAR-100 has $10\times$ more classes than the other 2 datasets, we increase proportionally both the # initial examples and the # examples per query so that the number of samples per class remains unchanged. We also increase the label budgets to a range of 4000 to 10000. Following Berthelot et al. (2019), we increase the size of the ResNet-28 model to 128 filters per layer for a more reasonable comparison.

Table 2 shows the results for CIFAR-100. Again, we found that active learning helps. The performance gain compared to the passive MixMatch ranges from 0.11% (4000 labels) to 1.16% (10000 labels). We found that methods with diversification (kmeans or infoD) consistently outperform their variants without diversification (direct). We conjecture that this is because given a large number of classes, diversifying the queried examples helps keep the sampled data class distribution more balanced. Besides, as in CIFAR-10, this is no significant difference between different active learning strategies, so we do not dismiss their effectiveness as well.

Table 2: Performance of MMA on 5 repeated runs on CIFAR-100.

| Label Budgets | 4000 | 5000 | 8000 | 10000 |
|---|---|---|---|---|
| MixMatch[1] | $67.72 \pm 0.79$ | $69.41 \pm 0.19$ | $72.87 \pm 0.32$ | $74.00 \pm 0.10$ |
| random | $67.56 \pm 0.50$ | $69.12 \pm 0.33$ | $72.27 \pm 0.28$ | $73.62 \pm 0.24$ |
| diff2.aug-direct | $67.72 \pm 0.50$ | $69.45 \pm 0.50$ | $72.80 \pm 0.28$ | $74.66 \pm 0.19$ |
| diff2.aug-kmeans | $67.82 \pm 0.48$ | $69.60 \pm 0.30$ | $73.16 \pm 0.29$ | $75.10 \pm 0.12$ |

**SVHN.** We evaluate MixMatch and variants of MMA with diff2 as uncertainty measurement on SVHN with 500, 1000, 2000 and 4000 labeling budget. We first train the model for 131072 steps with 250 randomly selected labeled samples. Then we grow the labeled set by 50 examples each time and continue training the model for 16384 steps. An adjustment made to MMA here is that we select the initial set of samples such that the class distribution is the same as that of the whole training set. This is because unlike CIFAR-10 and CIFAR-100, SVHN has unbalanced classes, where the largest class has 2.8 times more samples than the smallest. We note that MixMatch (Berthelot et al., 2019) selects the labeled samples in the same way.

Table 3 shows the results for SVHN. MMA outperforms MixMatch in all labeling budgets. MMA with 4000 examples is only 0.02% less accurate (97.39% vs. 97.41% according to Berthelot et al. (2019)) than the same model trained on all the 73257 examples ($18.31\times$ more) in the training set. We also found that diversification is useful, with kmeans method performing the best. We believe that diversification helps prevent under-sampling less popular classes, which can be beneficial when the classes are unbalanced.

---

[1]The discrepancy between the value for MixMatch and the values in Berthelot et al. (2019) for 10000 labels (which is $74.12 \pm 0.30$) can be explained by the fact that MixMatch selects samples uniformly at random while Berthelot et al. (2019) samples the same number from each class.

Table 3: Performance of `MMA` on 5 repeated runs on SVHN (no extra data).

| Label Budgets | 500 | 1000 | 2000 | 4000 |
|---|---|---|---|---|
| `MixMatch` | $96.36 \pm 0.46$ | $96.73 \pm 0.31$ | $96.96 \pm 0.13$ | $97.11 \pm 0.06$ |
| `random` | $96.44 \pm 0.18$ | $96.73 \pm 0.11$ | $96.83 \pm 0.07$ | $97.01 \pm 0.11$ |
| `diff2.aug-direct` | $96.59 \pm 0.09$ | $96.82 \pm 0.08$ | $97.03 \pm 0.05$ | $97.32 \pm 0.03$ |
| `diff2.aug-kmeans` | $96.69 \pm 0.07$ | $96.90 \pm 0.08$ | $97.14 \pm 0.06$ | $97.39 \pm 0.08$ |

### 5.3 COMPARATIVE ADVANTAGE OF ADDING LABELED VS. UNLABELED DATA

In this section, we compare the accuracy improvements offered by adding labeled data and unlabeled data. As proposed in Section 4.3, we measured the accuracy obtained by our model for various amounts of labeled and unlabeled data (see tables in Appendix A.2). To perform the analysis, we need to estimate the number of unlabeled samples $u_i$ required to achieve a $target_{accuracy}$ for a fixed amount of labeled data. For practical computational constraints, we approximate this process by using a linear interpolation between consecutive accuracy measurements when growing the amount of unlabeled data for a fixed amount of labeled data (see Appendix A.3 for details).

We plot the cost ratio $c_{ratio(i)}$ as a function of the number of labeled examples for CIFAR-10 and SVHN+Extra in Figure 1. For example, we can see that for CIFAR-10 with 500 labeled samples, it is cost efficient to label an image when the cost of labeling an image is less than 20 times the cost of obtaining new unlabeled image. We make the following observations:

- The more labeled samples are present, the more the value of labeling additional samples drops.
- Intriguingly, for SVHN, the ratio drops below 0 in some cases. This means that it is always better to add unlabeled data regardless of the labeling cost. This usually occurs when the assumption $|L| \ll |U|$ is violated, while MixMatch is tuned to operate under this assumption. Re-tuning of the parameters may be necessary to counter this effect.
- Generally, the highest cost ratio we observed was $20\times$ and frequently less than $3\times$ which makes labeling data a costly alternative with sample efficient techniques such as MixMatch.
- Likely, there exists a critical mass below which labeled data is always needed, but we did not observe it in our experiments. We conjecture that this number is below 50 samples per class.

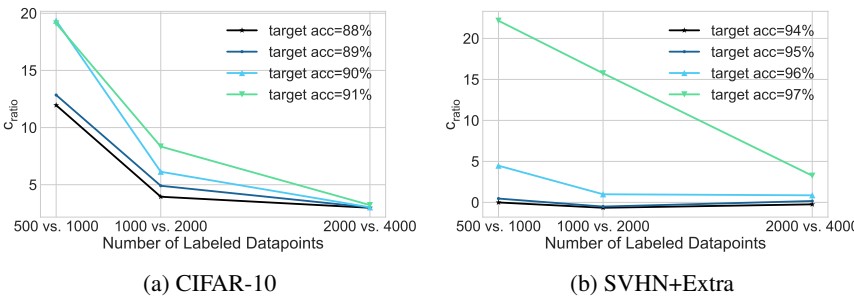

(a) CIFAR-10  (b) SVHN+Extra

Figure 1: Plot of $c_{ratio}$ as a measure of relative cost of labeling vs. obtaining unlabeled data.

### 6 CONCLUSION

In this work we have demonstrated that the state-of-the-art SSL MixMatch algorithm can be significantly improved when combined with various active learning methods, pushing the state-of-the-art even further. We found that generally, uncertainty sampling (via the `diff2` definition) performs robustly across several baseline datasets. We furthermore, provided an analysis to measure the relative value of labeled an unlabeled data and, interestingly, see that in certain regimes labeled data provide only a small constant factor (i.e. $< 3x$) additional value of unlabeled data. Future directions for this line of work include evaluating a trade-off aware algorithm, which dynamically adjusts the number of labeled and unlabeled examples that are gathered given the costs of each.

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

## A APPENDIX

### A.1 HYPER-PARAMETERS

Table 4 shows the hyper-parameters used throughout the paper.

Table 4: A summary of hyper-parameters. $\lambda_U$ is the hyper-parameter controlling the importance of the unlabeled data to the training process.

|  | CIFAR-10 | CIFAR-100 | SVHN | SVHN+Extra |
|---|---|---|---|---|
| # filters | 32 | 128 | 32 | 32 |
| $\lambda_U$ | 75 | 150 | 250 | 250 |
| MixUp | 0.75 | 0.75 | 0.75 | 0.25 |
| weight decay | 0.02 | 0.04 | 0.02 | 0.0001 |

### A.2 FULL RESULTS

#### A.2.1 EFFECTIVENESS OF MMA

Table 5, 6, 8 are the full versions of Table 1, 2, 3 respectively, presenting the results of different MMA variants for CIFAR-10, CIFAR-100 and SVHN.

Figure 2 show the plots of accuracy achieved versus the number of labeled samples for CIFAR-10, CIFAR-100 and SVHN.

Table 5: This is the full version of Tabel 1. Performance of `MMA` on CIFAR-10. Each entry is the average of 5 repeated runs with standard deviation. The highlighted entries are the two best performing methods within each column. The best methods for 500 is `max-kmeans` and for the rest is `diff2.aug-direct`.

| Label Budgets | 500 | 1000 | 2000 | 4000 | 16000 |
|---|---|---|---|---|---|
| Supervised | $60.72 \pm 1.04$ | $71.64 \pm 0.80$ | $80.26 \pm 0.76$ | $86.37 \pm 0.19$ | - |
| MeanTeacher | $61.95 \pm 6.80$ | $81.57 \pm 2.49$ | $87.28 \pm 0.84$ | $89.37 \pm 0.14$ | - |
| VAT | $73.69 \pm 2.34$ | $81.12 \pm 0.76$ | $85.88 \pm 0.47$ | $88.56 \pm 0.24$ | - |
| PiModel | $56.67 \pm 1.63$ | $68.77 \pm 0.82$ | $77.29 \pm 0.39$ | $84.23 \pm 0.80$ | - |
| PseudoLabel | $57.56 \pm 1.03$ | $68.96 \pm 1.66$ | $77.94 \pm 0.55$ | $83.84 \pm 0.28$ | - |
| MixMatch | $90.58 \pm 0.83$ | $91.61 \pm 0.54$ | $93.20 \pm 0.11$ | $93.70 \pm 0.16$ | $94.99 \pm 0.15$ |
| random | $90.54 \pm 0.82$ | $91.85 \pm 0.67$ | $92.97 \pm 0.29$ | $93.73 \pm 0.18$ | - |
| diff2.aug-direct | $\mathbf{91.69 \pm 0.52}$ | $\mathbf{92.79 \pm 0.41}$ | $\mathbf{94.11 \pm 0.14}$ | $\mathbf{95.17 \pm 0.13}$ | - |
| diff2-direct | $91.32 \pm 1.24$ | $\mathbf{92.89 \pm 0.24}$ | $93.97 \pm 0.15$ | $95.04 \pm 0.23$ | - |
| diff2.aug-kmeans | $91.46 \pm 0.38$ | $92.62 \pm 0.32$ | $93.98 \pm 0.14$ | $95.06 \pm 0.19$ | - |
| diff2-kmeans | $91.24 \pm 0.71$ | $92.66 \pm 0.62$ | $93.99 \pm 0.09$ | $95.09 \pm 0.11$ | - |
| diff2.aug-infoD | $91.52 \pm 0.67$ | $92.78 \pm 0.28$ | $93.97 \pm 0.09$ | $\mathbf{95.12 \pm 0.13}$ | - |
| diff2-infoD | $90.18 \pm 1.90$ | $92.76 \pm 0.25$ | $\mathbf{94.07 \pm 0.18}$ | $95.10 \pm 0.11$ | - |
| max.aug-direct | $90.87 \pm 1.65$ | $92.12 \pm 0.44$ | $93.72 \pm 0.22$ | $94.98 \pm 0.10$ | - |
| max-direct | $91.02 \pm 0.55$ | $92.09 \pm 0.51$ | $93.62 \pm 0.20$ | $95.03 \pm 0.05$ | - |
| max.aug-kmeans | $91.05 \pm 0.42$ | $92.06 \pm 0.50$ | $93.61 \pm 0.21$ | $94.90 \pm 0.21$ | - |
| max-kmeans | $\mathbf{91.78 \pm 0.42}$ | $92.55 \pm 0.46$ | $93.94 \pm 0.11$ | $95.00 \pm 0.09$ | - |
| max.aug-infoD | $90.94 \pm 0.59$ | $92.20 \pm 0.33$ | $93.55 \pm 0.36$ | $95.06 \pm 0.09$ | - |
| max-infoD | $90.60 \pm 0.82$ | $92.25 \pm 0.44$ | $93.75 \pm 0.22$ | $95.07 \pm 0.07$ | - |

Table 6: This is the full version of Tabel 2. Performance of `MMA` on CIFAR-100. Each entry is the average of 5 repeated runs with standard deviation. The highlighted entries are the two best performing methods within each column. Methods with diversification (`kmeans` or `infoD`) consistently outperform their variants without diversification (`direct`).

| Label Budgets | 4000 | 5000 | 8000 | 10000 |
|---|---|---|---|---|
| MixMatch[2] | $67.72 \pm 0.79$ | $69.41 \pm 0.19$ | $72.87 \pm 0.32$ | $74.00 \pm 0.10$ |
| random | $67.56 \pm 0.50$ | $69.12 \pm 0.33$ | $72.27 \pm 0.28$ | $73.62 \pm 0.24$ |
| diff2-direct | $67.37 \pm 0.26$ | $68.95 \pm 0.35$ | $72.61 \pm 0.18$ | $74.54 \pm 0.18$ |
| diff2.aug-direct | $67.72 \pm 0.50$ | $69.45 \pm 0.50$ | $72.80 \pm 0.28$ | $74.66 \pm 0.19$ |
| diff2-kmeans | $\mathbf{67.83 \pm 0.32}$ | $\mathbf{69.99 \pm 0.18}$ | $\mathbf{73.47 \pm 0.15}$ | $\mathbf{75.16 \pm 0.13}$ |
| diff2.aug-kmeans | $\mathbf{67.82 \pm 0.48}$ | $69.60 \pm 0.30$ | $73.16 \pm 0.29$ | $\mathbf{75.10 \pm 0.12}$ |
| diff2-infoD | $67.76 \pm 0.57$ | $\mathbf{69.84 \pm 0.16}$ | $\mathbf{73.32 \pm 0.23}$ | $75.03 \pm 0.22$ |
| diff2.aug-infoD | $67.78 \pm 0.61$ | $69.56 \pm 0.57$ | $72.95 \pm 0.33$ | $74.94 \pm 0.30$ |

## A.2.2 COMPARATIVE ADVANTAGE OF ADDING LABELED VS. UNLABELED DATA

## A.3 LINEAR INTERPOLATION COST RATIO MEASUREMENT

To efficiently find the number of unlabeled examples needed in order to reach a target accuracy at a fixed number of labeled training examples, consider the measurements in a table such as Table 9. For each column, we bisect the row values to find row $r$ such that $accuracy_{i,r} \leq accuracy_{target} \leq$

---

[2]The discrepancy between the value for `MixMatch` and the values in Berthelot et al. (2019) for 10000 labels (which is $74.12 \pm 0.30$) is due to the fact that `MixMatch` selects samples uniformly at random while Berthelot et al. (2019) samples the same number from each class. The difference becomes more significant when the number of classes grows.

[3]The 5 runs reaches accuracy $97.26, 97.39, 97.26, 93.42, 97.40$ and the variance is due to the oscillations in the training process.

Table 7: CIFAR-100. Starting from 2500 random samples trained for 262144 steps, querying 100 and training for 16384 steps each time. Each entry is the average of 5 repeated runs with standard deviation. The highlighted entries are the two best performing methods within each column. The accuracy is not significant different from that when query 500 each time (Table 6).

| Label Budgets | 4000 | 5000 | 8000 | 10000 |
|---|---|---|---|---|
| MixMatch | $67.72 \pm 0.79$ | $69.41 \pm 0.19$ | $72.87 \pm 0.32$ | $74.00 \pm 0.10$ |
| random | $66.84 \pm 0.43$ | $68.72 \pm 0.34$ | $72.18 \pm 0.31$ | $73.70 \pm 0.41$ |
| diff2-direct | $67.47 \pm 0.29$ | $69.25 \pm 0.27$ | $72.79 \pm 0.26$ | $74.68 \pm 0.33$ |
| diff2.aug-direct | $67.46 \pm 0.43$ | $69.35 \pm 0.37$ | $72.91 \pm 0.25$ | $74.50 \pm 0.35$ |
| diff2-kmeans | $67.89 \pm 0.27$ | $69.83 \pm 0.45$ | $73.10 \pm 0.33$ | $74.80 \pm 0.13$ |
| diff2.aug-kmeans | $67.64 \pm 0.61$ | $67.78 \pm 3.79$ | $73.07 \pm 0.20$ | $74.82 \pm 0.26$ |
| diff2-infoD | $\mathbf{68.06 \pm 0.58}$ | $\mathbf{70.18 \pm 0.52}$ | $\mathbf{73.90 \pm 0.21}$ | $\mathbf{75.26 \pm 0.23}$ |
| diff2.aug-infoD | $\mathbf{68.00 \pm 0.76}$ | $\mathbf{70.04 \pm 0.41}$ | $73.70 \pm 0.30$ | $\mathbf{75.45 \pm 0.27}$ |

Table 8: This is the full version of Tabel 3. Performance of MMA on SVHN (no extra data). Each entry is the average of 5 repeated runs with standard deviation. The highlighted entries are the two best performing methods within each column. Methods with kmeans diversification generally outperforms the other methods.

| Label Budgets | 500 | 1000 | 2000 | 4000 |
|---|---|---|---|---|
| MixMatch | $96.36 \pm 0.46$ | $96.73 \pm 0.31$ | $96.96 \pm 0.13$ | $97.11 \pm 0.06$ |
| random | $96.44 \pm 0.18$ | $96.73 \pm 0.11$ | $96.83 \pm 0.07$ | $97.01 \pm 0.11$ |
| diff2.aug-direct | $\mathbf{96.59 \pm 0.09}$ | $96.82 \pm 0.08$ | $97.03 \pm 0.05$ | $97.32 \pm 0.03$ |
| diff2-direct | $96.56 \pm 0.15$ | $96.85 \pm 0.03$ | $97.01 \pm 0.06$ | $97.33 \pm 0.03$ |
| diff2-kmeans | $\mathbf{96.59 \pm 0.06}$ | $\mathbf{96.91 \pm 0.07}$ | $\mathbf{97.09 \pm 0.04}$ | $97.36 \pm 0.03$ |
| diff2.aug-kmeans | $\mathbf{96.69 \pm 0.07}$ | $\mathbf{96.90 \pm 0.08}$ | $\mathbf{97.14 \pm 0.06}$ | $\mathbf{97.39 \pm 0.08}$ |
| diff2-infoD | $96.51 \pm 0.13$ | $96.76 \pm 0.12$ | $97.02 \pm 0.03$ | $97.29 \pm 0.05$ |
| diff2.aug-infoD | $96.51 \pm 0.15$ | $96.73 \pm 0.10$ | $96.28 \pm 1.60$ | $\mathbf{97.37 \pm 0.04}$ |

$accuracy_{i,r+1}$. We then compute the interpolation factor

$$\lambda_i = \frac{accuracy_{target} - accuracy_{i,r+1}}{accuracy_{i,r} - accuracy_{i,r+1}}.$$

Finally the number of unlabeled samples can be linearly approximated as

$$u_i = \lambda_i \cdot accuracy_{i,r} + (1 - \lambda_i) \cdot accuracy_{i,r+1}.$$

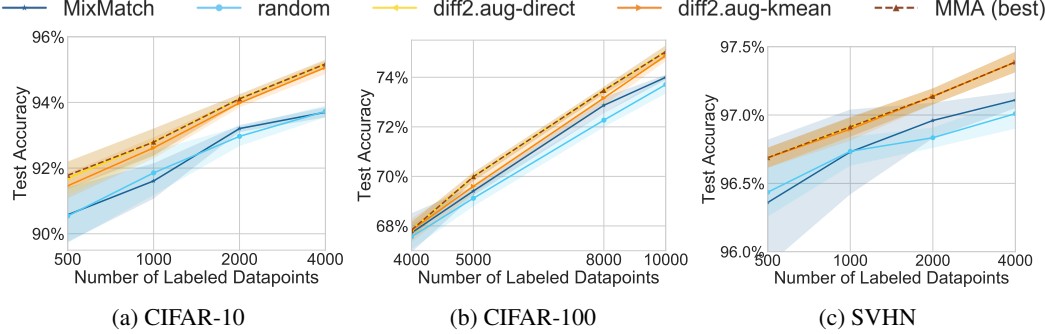

Figure 2: Test accuracy (y-axis) under different number of labeled samples (x-axis, log scale). MMA (best) refers to the highest accuracy that any MMA variants achieves (with full results listed in tables found in Appendix A.2). Note, the AL methods are essentially overlapping.

Table 9: CIFAR-10. Accuracy of `MMA` with `diff2.aug-direct` using different amounts of labeled and unlabeled data. The first column indicates the total number of datapoints (labeled + unlabeled).

| | Labeled | | | |
|---|---|---|---|---|
| Total | 500 | 1000 | 2000 | 4000 |
| 5000 | $64.45 \pm 0.98$ | $66.69 \pm 1.15$ | $71.71 \pm 1.27$ | $80.33 \pm 0.71$ |
| 10000 | $72.79 \pm 2.02$ | $74.81 \pm 1.57$ | $78.78 \pm 0.79$ | $84.65 \pm 0.36$ |
| 15000 | $77.65 \pm 2.13$ | $79.57 \pm 1.93$ | $83.26 \pm 1.09$ | $87.87 \pm 0.62$ |
| 20000 | $80.80 \pm 1.48$ | $83.16 \pm 1.47$ | $87.20 \pm 1.31$ | $90.98 \pm 0.52$ |
| 25000 | $85.29 \pm 1.24$ | $87.96 \pm 0.67$ | $90.40 \pm 0.53$ | $93.12 \pm 0.22$ |
| 30000 | $87.61 \pm 0.59$ | $89.89 \pm 0.79$ | $92.31 \pm 0.32$ | $93.95 \pm 0.19$ |
| 35000 | $89.32 \pm 0.60$ | $91.04 \pm 0.67$ | $93.10 \pm 0.28$ | $94.55 \pm 0.12$ |
| 40000 | $89.97 \pm 0.99$ | $91.83 \pm 0.70$ | $93.37 \pm 0.11$ | $94.59 \pm 0.15$ |
| 45000 | $91.14 \pm 0.46$ | $92.60 \pm 0.30$ | $94.05 \pm 0.12$ | $95.08 \pm 0.04$ |
| 50000 | $91.69 \pm 0.52$ | $92.79 \pm 0.41$ | $94.11 \pm 0.14$ | $95.17 \pm 0.13$ |

Table 10: CIFAR-100. `MMA` with `diff2.aug-kmeans`. Values at $(5000, 5000)$ and $(10000, 10000)$ are computed with fully supervised learning on the same model.

| | Labeled | | | |
|---|---|---|---|---|
| Total | 4000 | 5000 | 8000 | 10000 |
| 5000 | $48.40 \pm 0.50$ | $55.42 \pm 0.30$ | - | - |
| 10000 | $53.12 \pm 0.55$ | $55.39 \pm 0.56$ | $60.21 \pm 0.27$ | $65.93 \pm 0.34$ |
| 20000 | $59.78 \pm 0.21$ | $61.67 \pm 0.27$ | $65.11 \pm 0.11$ | $67.01 \pm 0.21$ |
| 50000 | $67.82 \pm 0.48$ | $69.60 \pm 0.30$ | $73.16 \pm 0.29$ | $75.10 \pm 0.12$ |

Table 11: SVHN+Extra. `MMA` with `diff2.aug-kmeans`.

| | Labeled | | | |
|---|---|---|---|---|
| Total | 500 | 1000 | 2000 | 4000 |
| 5000 | $91.04 \pm 0.27$ | $90.93 \pm 0.30$ | $90.50 \pm 0.42$ | $89.66 \pm 0.25$ |
| 10000 | $93.71 \pm 0.13$ | $93.70 \pm 0.17$ | $93.54 \pm 0.14$ | $93.35 \pm 0.19$ |
| 20000 | $95.28 \pm 0.23$ | $95.32 \pm 0.22$ | $95.27 \pm 0.21$ | $95.37 \pm 0.19$ |
| 50000 | $96.48 \pm 0.06$ | $96.61 \pm 0.08$ | $96.75 \pm 0.09$ | $96.83 \pm 0.07$ |
| 100000 | $97.07 \pm 0.07$ | $97.17 \pm 0.04$ | $97.40 \pm 0.07$ | $97.56 \pm 0.08$ |
| 200000 | [3] $96.55 \pm 1.56$ | $97.44 \pm 0.08$ | $97.66 \pm 0.04$ | $97.87 \pm 0.04$ |
| 604388 | $97.38 \pm 0.19$ | $97.53 \pm 0.18$ | $97.80 \pm 0.08$ | $97.98 \pm 0.05$ |

