# OpenReview forum: "Combining MixMatch and Active Learning for Better Accuracy with Fewer Labels"
_ICLR.cc/2020/Conference — Reject_

### Official Review · AnonReviewer3 · 2019-10-21
**Official Blind Review #3**

**Rating:** 6

**Review:**

This paper takes a look at using active learning techniques instead of random sampling for the
"state-of-the-art" semi-supervised learning (SSL) method MixMatch. At least for this one SSL algorithm, the authors give a strong argument that active learning helps MixMatch (4x label efficiency in some cases) and highlight the active learning algorithms that work best. An additionally interesting point is the value of labeled vs unlabeled data in this setting. For these reasons, I argue for acceptance of this paper. However, I have some reservations which are given below, that perhaps the authors can clarify.


Things that would have improved my score:

 - This paper relies on MixMatch very heavily as the sole semi-supervised technique. It would be nice to see more of an argument for this choice of a relatively recent paper that hasn't stood the test of time.

 - I am confused why the authors "report the median of the last 20 checkpoints' accuracy where a checkpoint is computed every 65,536 training iterations". As we see later, the authors train after each batch of selected examples for 32,768 iterations which is half of the time between checkpoints. Can the authors comment on this choice?


Minor comments:

 - The end of the abstract makes it sound like like the conclusions are universal ("quickly diminishes to less than 3x once more than 2000 labeled example are observed"). I would be surprised if the authors meant this as a universal statement since no theory is provided and the experiments are on similar datasets.

 - I don't understand why section 3 is not simply titled "MixMatch" since the paper doesn't really touch any other "modern SSL" methods.

 - In the experiments section, 262144 and 32768 iterations seem to come from nowhere. Only later did I realize that these were powers of 2. Can this be clarified?

 - What's the difference between MixMatch and MMA with random selection? Shouldn't these perform the same (which they seem to anyways)?

 - I really like that this paper assesses the label efficiency of their algorithm, rather than merely reporting raw accuracy numbers which aren't as meaningful.

 - I wonder if MMA seems to not give as big gains on CIFAR-100 because the batch size is 10x larger. Generally, I've found that active learning (especially uncertainty sampling methods) work best with smaller batch sizes and I'm not sure I agree with the reasoning that more classes mean one should select larger batch sizes.








**Experience Assessment:**

I have published one or two papers in this area.

**Review Assessment: Checking Correctness Of Derivations And Theory:**

I assessed the sensibility of the derivations and theory.

**Review Assessment: Checking Correctness Of Experiments:**

I assessed the sensibility of the experiments.

**Review Assessment: Thoroughness In Paper Reading:**

I read the paper at least twice and used my best judgement in assessing the paper.

---

> ### Author Response · Authors · 2019-11-14
> **Response to Review #3**
>
> Thank you very much for the careful reading and feedback. Here are our answers and clarifications.
>
> - MixMatch is the only SSL method evaluated.
> Please see response to Reviewer #2's question 2.
>
> - Question regarding checkpoints.
> Thank you very much for the careful reading and we apologize for the typo! Each checkpoint is saved every **2^10=1024** iterations (not every 2^16=65536 iterations).
>
> We use two parameters: the initial training duration before growing (262144=2^18 you quoted in your question) and the number of training steps between each repeated growth (32768=2^15 from your question).
> These numbers were chosen empirically to balance the convergence of the model and training time. The choice of power of 2 follows from the practice in the MixMatch work (https://github.com/google-research/mixmatch).
>
> - Universal statement at the end of the abstract.
> Indeed, we make these observations for our experiments; we will make this clear.
>
> - Section 3 title.
> We have modified this section to include discussion of other approaches as well.
>
> - Difference between MixMatch and MMA with random selection.
> MMA with random selection and MixMatch only differ in that MMA trains with an iteratively growing subset of (uniformly randomly selected) training data, while MixMatch trains on the entire training set at once. Thus, it is not surprising that the two methods perform similarly, although it is worthwhile to conduct the experiment since the two methods may converge differently given the non-convex nature of the training problem.
>
> - Performance on CIFAR-100 with smaller batch size.
> We ran a new set of experiment on CIFAR-100, with 2500 randomly selected samples initially and smaller batch size of 100 sampled label queries each time. We summarized the results here and also in the appendix (Table 7). Although some methods show improvement, they are within the standard deviation.
>                                 4000 		5000 		8000 		10000
> random 		        66.84±0.43 	68.72±0.34 	72.18±0.31 	73.70±0.41
> diff2-direct 		67.47±0.29 	69.25±0.27 	72.79±0.26 	74.68±0.33
> diff2.aug-direct 	67.46±0.43	69.35±0.37 	72.91±0.25 	74.50±0.35
> diff2.aug-kmean	67.64±0.61 	67.78±3.79 	73.07±0.20 	74.82±0.26
> diff2-kmean		67.89±0.27 	69.83±0.45 	73.10±0.33 	74.80±0.13
> diff2-id	   	  	68.06±0.58	70.18±0.52	73.90±0.21	75.26±0.23
> diff2.aug-id		68.00±0.76	70.04±0.41	73.70±0.30	75.45±0.27
>
> Hope these address your concerns.

---

### Official Review · AnonReviewer1 · 2019-10-22
**Official Blind Review #1**

**Rating:** 3

**Review:**

Summarize the paper:

This paper proposes a method that can deal with an active-learning scenario for the recently proposed semi-supervised learning method: MixMatch.  More specifically, the proposed method considers uncertainty measures to choose samples and a diversification step to ensure diversity within the sampled batch.  For uncertainty measures, the paper considers the simple maximum confidence and the gap between two most likely classes.  Additional augmentation techniques inspired from MixMatch are used.  For diversification, a clustering method and an information density method are considered.  Furthermore, the paper proposes a cost analysis model to compare labeled and unlabeled samples.  Experiments demonstrate the behavior of the proposed method.


Pros of the paper:

- The experimental results seem to be strong and encouraging.
- The discussions on the cost of labeled and unlabeled samples seems to be an important contribution for semi-supervised active learning.
- The motivation and direction of the paper is simple and easy to follow: Take the state-of-the-art semi-supervised learning algorithm and propose an active-learning version of it.
- It is not a straightforward combination of MixMatch and active learning, and there are some specialized techniques such as “aug” used in the design of the proposed algorithm.


Cons of the paper:

- Only uncertainty based sampling methods are considered, but is this enough?  There seems to be no other papers that deal with active semi-supervised learning for a deep learning context, so it might be important to really explore the many sampling methods (e.g., from survey of Settles 2009).

- A more minor comment: The same issue goes for the semi-supervised learning side.  MixMatch is the state of the art in terms of accuracy for image domains, but it is an ensemble of several semi-supervised learning methods, and have strong assumptions, e.g., smoothness assumption, small distribution overlap, etc.  This will mean the proposed method will also have those strong assumptions and limits the method’s applicability.

- In experiments, it would be better to have figures that are usually used in active learning experiments, where the x-axis is the remaining budget and y-axis is the performance measure.


Additional comments:

Active learning methods gives labels to unlabeled samples in different epochs until the budget is used up, but it would be interesting to give the final labeled and unlabeled dataset after budget is used up as a fixed dataset, and then train the traditional passive MixMatch with this.  Then we can really compare the original MixMatch and active MixMatch.  If the proposed method still works better,  then the proposed method might be meaningful not only as an active learning method but also as a curriculum learning method.

********************
I would like to thank the authors for answering my questions and updating the paper, but would like to keep my score due to the 2nd point of the cons.  A minor comment on the second point of the author response:  The sharpening step in MixMatch can be regarded as an entropy minimization procedure, which I think is based on the assumption of low distribution overlap.

**Experience Assessment:**

I have read many papers in this area.

**Review Assessment: Checking Correctness Of Derivations And Theory:**

I assessed the sensibility of the derivations and theory.

**Review Assessment: Checking Correctness Of Experiments:**

I assessed the sensibility of the experiments.

**Review Assessment: Thoroughness In Paper Reading:**

I read the paper at least twice and used my best judgement in assessing the paper.

---

> ### Author Response · Authors · 2019-11-14
> **Response to Review #1**
>
> Thank you very much for the careful reading and feedback! Here are our answers and clarifications.
>
> - "Only uncertainty based sampling methods are considered…"
> Although many other AL algorithms exist (such as those found in Settles 2009), we find that they are not easily adapted to the setting of complex neural networks which are costly to train. We have added an explanation at the end of Section 4.2.
>
> - "[MMA] will also have those strong assumptions…"
> The proposed method inherits the properties of the SSL algorithm being used. MixMatch mostly relies on consistency regularization and as far as we know does not make claims about distribution overlap.
>
> - "[Plot the figure so that] the x-axis is the remaining budget and y-axis is the performance measure"
> Figure 1 (now Figure 2 in the revision) is plotted in what we find is a common fashion (accuracy vs. # labeled examples). Is the main comment that you would prefer to see the same plot but with the x-axis reversed (i.e., budget remaining rather than budget consumed)? This can easily be done the camera ready version.
>
> Hope these address your concerns.

---

### Official Review · AnonReviewer2 · 2019-10-23
**Official Blind Review #2**

**Rating:** 3

**Review:**

The paper proposes to combine active learning techniques with MixMatch for semi-supervised learning. First, they review active learning and semi-supervised learning, especially MixMatch. Instead of traditional semi-supervised learning with a fixed set of labeled examples, they incrementally grow the labeled set as the training process goes on. They consider several different choices in active learning strategies: uncertainty measure and diversification. Diversification methods are used to balance the samples in different classes and ensure diversity. The cost analysis of adding labeled vs unlabeled data looks interesting. They perform an empirical evaluation on image benchmarks and improve over MixMatch.

Overall, the paper is clearly written and easy to follow. However,  I cannot recommend acceptance because

1. Novelty concern. The combination of two existing techniques seems not novel enough.

2. Missing important baselines in related work and experiments. In the related work on semi-supervised learning (Section 3), the authors only review MixMatch but neglect other literature, e.g.[1,2,3,4]. And semi-supervised learning has a long history and it is not restricted to recent deep learning-based approaches. A thorough review can make the approach well-placed in the literature. In experiments, the authors only compare with MixMatch. I suggest that the authors include the missing literature in the next version.

3. The cost analysis is the most interesting to me. However, Figure 2(b) in Section 5.3 is weird. How can the ratio less than 0? According to the definition in Section 4.3, $L_i \subset L_{i+1}$ and $|L_{i+1}| > |L_i|$ and similar case for $U_{i,j}$, the cost ratio should not be less than 0. I'm also confused by the explanation in Section 5.3.

4. From Figure 1(b) and Table 2, we can see that on CIFAR-100, the improvement of the proposed MMA is not statistically significant, especially when the label budget is low. But on a simpler dataset CIFAR-10, MMA performs better. How does MMA perform on a more challenging task with more classes, e.g. ImageNet?

5. Training time comparison. At the expense of spending more time on selecting uncertain examples and techniques like k-means clustering, MMA is slightly better than MixMatch. A comparison of training time and complexity would be better to convince me.



***
Minor:
page 4 “Starting with from a fixed pool of n unlabeled sample”
page 4 “A corollary question is how do various accuracy targets relate to each other?”
page 5 “While there are there additional active learning”
page 5 “ Let’s define cl and cu as the cost of respectively obtaining a new labeled sample and a new unlabeled sample.” --> costs



***
References
[1] Temporal Ensembling for Semi-Supervised Learning, ICLR 2017.
[2] Smooth Neighbors on Teacher Graphs for Semi-supervised Learning, CVPR 2018.
[3] Realistic Evaluation of Semi-supervised Learning Algorithms, NeurIPS 2018.
[4] There Are Many Consistent Explanations of Unlabeled Data: Why You Should Average, ICLR 2019.



**Experience Assessment:**

I have published one or two papers in this area.

**Review Assessment: Checking Correctness Of Derivations And Theory:**

I carefully checked the derivations and theory.

**Review Assessment: Checking Correctness Of Experiments:**

I carefully checked the experiments.

**Review Assessment: Thoroughness In Paper Reading:**

I read the paper thoroughly.

---

> ### Author Response · Authors · 2019-11-14
> **Response to Review #2**
>
> Thank you very much for the careful reading and feedback. Here are our answers and clarifications.
>
> 1. Novelty concern
> We make two points to help answer this concern. Even if the two methods (MixMatch and uncertainty based AL) are themselves not novel, their combination is and the fact that they result in state-of-the-art performance, we believe, is a worthwhile discovery. Secondly, we do note that the combination is not entirely straightforward: for example, we borrowed the idea of using multiple augmentations from MixMatch when we compute the confidence scores of unlabeled samples (namely “aug”). An “easy” sample should mean that the current model can predict confidently on multiple augmentations of it, and these predictions are consistent. This can potentially improve the stability of our confidence estimation. Empirically, we can also see this method helps.
>
> 2. SSL baselies other than MixMatch.
> We have added references to several other SSL methods to the beginning of Section 3. Furthermore we have added the following empirical evaluations:
>
> We run four other SSL methods on CIFAR-10 in the passive setting and report the mean and standard deviation over 5 repeated runs are summarized here, together with that for MixMatch.
>
> 			      500		        1000		  2000		    4000
> MeanTeacher     61.95±6.80       81.57±2.49       87.28±0.84       89.37±0.14
> VAT         	     73.69±2.34     	81.12±0.76       85.88±0.47       88.56±0.24
> PiModel     	     56.67±1.63     	68.77±0.82       77.29±0.39	    84.23±0.80
> PseudoLabel      57.56±1.03     	68.96±1.66       77.94±0.55       83.84±0.28
> MixMatch	     90.58±0.83        91.61±0.54	  93.20±0.11	    93.70±0.16
>
> We chose MixMatch to augment with active learning since its SSL performance is significantly superior. The MixMatch work[1] confirms the performance of MixMatch compared to other SSL methods, showing that MixMatch outperforms other SSL methods by a large margin in different datasets.
> We added the performance measurements of the above SSL methods in the appendix (Table 5) to justify the choice of MixMatch, and we added a brief summary of popular SSL methods in Section 3.
> [1] Berthelot, David, et al. "Mixmatch: A holistic approach to semi-supervised learning." arXiv preprint arXiv:1905.02249 (2019).
>
> 3. Cost ratio less than 0.
> The cost ratio can be less than zero when the difference |U_{i,j}| - |U_{i+1, k}| < 0, which can for example occur when adding labeled data actually causes a drop in accuracy. This occurs in Table 10 when the number of unlabeled points is small. This is likely due to the fact that MixMatch algorithm has been tuned to work best in the regime when |U| >> |L|.
>
> 4. Improvement on CIFAR-100 not statistically significant.
> Indeed, the improvement due to MMA is smaller on CIFAR-100 compared to other datasets. This is likely related to the general observation that active learning provides less of a benefit as a task become more difficult. For example see:
>
> Mussmann, Stephen, and Percy Liang. "On the Relationship between Data Efficiency and Error for Uncertainty Sampling." International Conference on Machine Learning. 2018.
>
> Finding additional techniques to complement active learning in these cases is an interesting direction, however, is outside the scope of this investigation.
>
> We did not run on ImageNet due to computational constraints at the time of writing and the lack of availability of MixMatch parameters for this dataset. SSL publications only recently started to study ImageNet (in the last 6 months).
>
>
> 5. Training time comparison.
> In general, the cost of the AL algorithm is dwarfed by the training time of the machine learning model which, in real-world applications, is itself dwarfed by the time it takes to gather labels for queried points. Thus, the additional cost introduced by active learning is generally insignificant. However, to give an idea, we measured the time (in seconds) it takes per active learning iteration to decide which unlabeled samples to query with three AL methods for CIFAR-10 (w/ 49750 unlabeled samples to choose from), CIFAR-100 (w/ 47500 unlabeled) and SVHN (w/ 73007 unlabeled). As can be seen in the table, the time for deciding the unlabeled data to query is usually less than 5 minutes. Compared to the model training time, which, for a passive MixMatch to be trained until convergence, is ~15h for CIFAR-10, ~24h for CIFAR-100 and ~8h for SVHN, the time for this decision process is not significant.
>
> 		        diff2-direct	diff2-kmean	diff2-id
> CIFAR-10	5.78±1.56	20.67±6.69	100.06±18.20
> CIFAR-100	21.62±2.47	81.92±18.63	175.06±22.60
> SVHN		13.04±1.00	37.84±14.75	191.87±22.95
>
> Hope these address your concerns.

---

### Decision · Program_Chairs · 2019-12-19

**Decision:**

Reject

**Comment:**

This paper extends state of the art semi-supervised learning techniques (i.e., MixMatch) to collect new data adaptively and studies the benefit of getting new labels versus adding more unlabeled data. Active learning is incorporated in a natural and simple (albeit, unsurprising) way and the experiments are convincing that this approach has merit.

While the approach works, reviewers were concerned about the novelty of the combination given that its somewhat obvious and straightforward to accomplish. Reviewers were also concerned that the space of both semi-supervised learning algorithms and active learning algorithms was not sufficiently exhaustively studied. As one reviewer points out: neither of these ideas are new or particular to deep learning.

Due to lack of novelty, this paper is not suited for a top tier conference.